# Effects of Climate Change and Maternal Morality: Perspective from Case Studies in the Rural Area of Bangladesh

**DOI:** 10.3390/ijerph16234594

**Published:** 2019-11-20

**Authors:** Abu Sayeed Md. Abdullah, Koustuv Dalal, Abdul Halim, AKM Fazlur Rahman, Animesh Biswas

**Affiliations:** 1Centre for Injury Prevention and Research Bangladesh (CIPRB), Dhaka 1206, Bangladesh; sayeedvetmicro@gmail.com (A.S.M.A.); halim.ogsb@gmail.com (A.H.); fazlur@ciprb.org (A.F.R.); ani72001@gmail.com (A.B.); 2School of Health Sciences, Mid Sweden University, 851 70 Sundsvall, Sweden; 3Institute of Health Sciences, Örebro University, 701 82 Örebro, Sweden

**Keywords:** maternal death, marginalized community, flood, natural disaster, Bangladesh

## Abstract

This study explored the community perception of maternal deaths influenced by natural disaster (flood), and the practice of maternal complications during natural disaster among the rural population in Bangladesh. It also explored the challenges faced by the community for providing healthcare and referring the pregnant women experiencing complications during flood disaster. Three focus group discussions (FGDs) and eight in-depth interviews (IDIs) were conducted in the marginalized rural communities in the flood-prone Khaliajhuri sub-district, Netrakona district, Bangladesh. Flood is one of the major risk factors for influencing maternal death. Pregnant women seriously suffer from maternal complications, lack of antenatal checkup, and lack of doctors during flooding. During the time of delivery, it is difficult to find a skilled attendant, and referring the patient with delivery complications to the healthcare facility. Boats are the only mode of transport. The majority of maternal deaths occur on the boats during transfer from the community to the hospital. Rural people feel that the maternal deaths influenced by natural disaster are natural phenomena. Pre-preparation is needed to support pregnant women during disasters. There is unawareness of maternal health, related care, and complications during disasters among local health service providers and volunteers.

## 1. Introduction

In rural Bangladesh, natural disasters are identified as one of the most important factors for deaths of women, especially during pregnancy. The female to male death ratio is 3:1 during natural disasters [1]. During the disaster, women are the most vulnerable for pregnancy complications including retained placenta, obstructed labor, and fetal distress. It is difficult to manage maternal health problems during disasters as healthcare facilities and providers are not available. Delivery in unsafe condition increases maternal deaths [2]. Disaster also impacts on reproductive health through spontaneous abortion, birth defects, and low birth weight of the baby [2,3]. According to literature, South-East Asia is the most vulnerable region, due to global warming having significant effects on the climate changes including unprecedented heavy rain and floods [4,5,6,7]. In Pakistan, about 500,000 expecting mothers were affected by the 2010 flood while 1.5 million women needed emergency obstetric care [4]. Among pregnant women during the disaster, 1700 women delivered, with hundreds of them suffering from delivery complications in Pakistan. Maternal deaths are also high in the Indian sub-continent due to a lack of medicine and the absence of female healthcare providers during disasters [5]. Bangladesh is considered to be one of the world’s most natural hazard-prone countries, and flood is one of the most common disasters experienced regularly by the people of Bangladesh where on average, 18% of the country is affected by floods every year [6]. In a study by UNFPA of nine districts in Bangladesh, it was found that 1,876,636 people were affected by flood disasters, of which 32,000–33,000 were pregnant women [7]. Pregnant women, lactating mothers, and differently disabled women suffered the most, as they found it difficult to move during and after a disaster. Sometimes women cannot express their problems [8]. In Bangladesh, it is found that non-availability of transport in and around all flood-affected areas and disruption of communications seriously hindered women’s ability to access health facilities for deliveries [9]. Moreover, the delay in decision making and delays in transportation influence the maternal deaths in rural communities [10]. It is also found that some of the healthcare centers are inundated with flood water. So, access to health services becomes limited as a result of routine immunization and outpatient consultation, antenatal care become disrupted in the affected villages in Bangladesh [11]. Due to climate change, Bangladesh is overexposed for natural disasters such as floods. This study explored the perception of maternal death during the flood. This study also explored the practices and challenges of the community people for emergency maternal care with complications during the flood period. The study tried to investigate the community recommendation for preventing maternal death during flood. 

## 2. Materials and Methods

### 2.1. Study Methods

A qualitative study was conducted at Khaliajhuri Upazila (sub-district) in Netrakona district of Bangladesh during July to September 2015. The sub district is affected by floods every year, and boats are the only mode of transportation during rainy seasons and floods. 

Three focus group discussions (FGD) and eight in-depth interviews (IDI) were conducted in two unions of the sub-district where three maternal deaths occurred during the previous flood. For FGDs, we chose three groups. The participants who were included in the FGDs were people who knew details about the respective maternal deaths. Each group was selected from the union where the maternal deaths occurred during the previous flood. FGD members consisted of the neighbors of the deceased mother’s family, male and female guardians of pregnant and recently delivered mothers, pregnant women, community group members, school teachers, religious leader, Union Parisad members, and elite people of the society who have idea on the incidence of the maternal death. Nine to eleven participants were included in each of the FGD.

For in-depth interviews, participants were chosen from specific communities (where FGDs were conducted). Eight IDIs were performed; Two were conducted with the male guardian and two were conducted with the female guardians of the pregnant or recently delivered mother, two were conducted with the village doctors, and the remaining two were conducted with the traditional birth attendant of that specific community where maternal death was reported. All IDIs were conducted following guidelines by face-to-face interview at the household level. This qualitative research technique has vast advantages to explore the interviewee’s perspective on a particular situation [12,13]. Time duration for FGD conduction was 30–40 minutes, whereas for the IDI it was 15–20 minutes (Table 1).

### 2.2. Data Collection

Field training was conducted among two research officers and guidelines were pre-tested. One research officer was a trained and experienced anthropologist (postgraduate), who performed as a moderator, and the other research officer was from social science background, who performed as a note taker). During FGDs, one research officer facilitated the discussion, while the other research officer took important notes. The objectives of the research were demonstrated to the respondents before the interviews. A written consent was taken from each of the respondents before the interviews or FGDs. A number of prompts were used to obtain the information. Audio voice recording was done with prior permission from the respondents. From the audio-recordings and hand notes of the interviewers, the research officers prepared verbatim transcripts of the IDIs and FGDs in native Bengali language. Later, English translations of the transcripts were performed by two expert bi-lingual researchers. The principal researcher controlled the transcript quality by randomly selected transcripts reviews and translation. 

These transcriptions were also checked by public health specialists. Peer debriefing was also performed to maintain the reliability of the data. Initial open coding was done, then from these open codes, selective coding was done. Contents were identified after read and re-read of the data [14,15] and finally, content analysis was performed (Table 2). 

### 2.3. Data Analysis

Qualitative content analysis was conducted following the guidelines by Graneheim and Lundman [16]. The participants’ words were analyzed as actual content, and interpretation and judgment of participants’ response was analyzed as latent content [17]. We analyzed the data with a repeated look over the written transcription by identifying each of the meaning units and listening to the audio recorder [16]. 

## 3. Results

Flood is a common and annual natural disaster in the study area, especially during monsoon in the Khaliajhuri sub-district of the Netrakona district, Bangladesh. Maternal healthcare is seriously disrupted during such a disaster period. There is community ignorance on specialized maternal care as a whole, including during disaster period. Healthcare providers are not available during maternal complications of the mother during and after the disaster. It is difficult to organize delivery of the mother during and after the disaster period. It is a very complex task to organize a place and person for delivery. A skilled birth attendant for delivery are frequently unavailable in the disaster area. It is even very difficult to organize as traditional birth attendant for assisting the delivery. If delivery complication arises, then referral of the mother from the community to a facility is a very difficult and cumbersome process. The boat is the most common mode of transportation. Transporting the risky mother with delivery complications to as healthcare facility takes a lot of time. As a result, sometimes mothers die on boats due to delays in transferring them to the hospital. 

### 3.1. Perception on Occurrence of Maternal Death during Flood 

Most of the community people who participated in FGD and IDI had a perception of natural disaster and maternal death in their area. However, they perceive that both of the things occur due to destiny/ill-fate, even though floods occur in one season (monsoon) but their effects persists throughout the year in that area. Floods make their life more complicated as there are problems of housing, communication, food, and medical treatment. Their lives become under threat during flooding. Pregnant women and their children are more vulnerable during floods. Maximum maternal death occurs during the disaster period due to difficulty in obtaining treatment and communication. Maternal deaths occur due to complications after delivery to delays in reaching the hospital due to communication and transportation problems. 

“*Flood occurs every year in rainy season but it was most dangerous in 2014. People couldn’t go out from home. Tube wells and toilets were sub-merged. We stayed on the roof of our houses. We could not cook due to wind and heavy rain. Many children died with diarrhea. Some people died by thunderstorms*”.—(P20, FGD 1) one of the male guardians mentioned

“*Maternal deaths occur due to excessive bleeding after delivery and difficult to reach at hospital as boat is the only mode of transportation. Sometimes the mother died within the boat. This is most common during the flood period*”.—(P13) village doctor

### 3.2. Practices of Maternal Healthcare during Natural Disaster

According to the community people, pregnant women suffer a lot during floods. They often have no prior planning for the management of maternal complications during floods. Pregnant women do not receive special attention in terms of care or treatment. Local public healthcare facilities and healthcare providers were not available for maternal care. The husband of a mother with severe labor pain commonly accompanied her during boat transportation. But if any complication arises, they communicate with the village doctors (rural medical practitioners without any formal medical degree), local kabiraj (a type of quack practicing traditional Ayurveda), and a traditional birth attendant for help. If any serious maternal complications arise, they arrange an emergency boat, trawler, or any available mode of transport to transfer the mother to the nearby hospital as per suggestion by the village doctor and traditional birth attendant. (N.B. Traditional Ayurveda is not a scientifically approved medicine. But people have practiced it for hundreds of years. It is locally made by traditional, untrained people. Kabiraj and village doctors are not government employed, rather they practice at rural areas without any professional skills).

“*We call kabiraj and village doctor if there is any complication of the pregnant mother during flood. They provide treatment after checkup. We can’t go to the public hospital due to difficulties in transportation. But if the kabiraj and village doctor fail to provide treatment and suggest us to go to the hospital then we try to arrange for transferring the mother which is very difficult. At first, we have to go to Krishnopur bazaar (nearby market place) to arrange a boat for reaching to Samachor (nearest available land with motorable road), then we book the auto-rickshaw or laguna (indigenously makeshift tri-cycle with installed water-pump motor) to reach hospital. The process is very time-consuming and expensive*”.—(P2, FGD 2)

*“Koki’s wife died with delivery complication because she was not sending to the hospital timely due to flood at that time in our village. When she reached the hospital she died. The road and transport system are not fine here and it’s difficult to reach hospital timely during emergency case which is a reason for the maternal death”*.—(P12, FGD 3)

“*I try to make delivery of the mother at home but during flood, it is difficult for me to reach the pregnant mother’s home. But if I realize that the mother is at risk then I immediately refer her to Upazila hospital though it is very difficult and time-consuming to reach the hospital by boat*”.—(IDI5) traditional birth attendant

### 3.3. Obstacle to Practices on Maternal Care during Disaster

Communication and transportation are major obstacles of maternal care in the flood-affected areas. During the flood, the public healthcare facilities or hospitals in rural areas are closed, and healthcare providers do not regularly checkup on pregnant women. Satellite clinics do not organize during the flood period. Though there are some volunteer activities for supporting flood-affected people, there is no special support for maternal healthcare. It is even difficult to find a village doctor or traditional birth attendant. The only vehicles are boats for transportation. When any difficulties arising among pregnant women, it is difficult to organize a boat. It is also very time-consuming to reach the hospitals. Sometimes people depend on destiny for any maternal complication during the flood, if they cannot organize any boat. Boatmen are not easily available, or they will not agree to carry the risk to transport the complicated pregnant mother. They demand a high amount of money for the emergency boat transportation.

“*Boat is the only vehicle for transportation during flood. Sometimes a boat can’t easily manage during an emergency. Boatmen demand a lot of money during transportation of mother with maternal complication as it is time-consuming*”.—(P8, FGD 1)

“*We can’t easily find any village doctor or traditional birth attendant during maternal complication at the period of flood as there is huge pressure of patient at that time. Sometimes we have nothing to do but just to depend only on the fate*”.—(IDI8) One of the female guardians

### 3.4. Barrier to Referral of Complicated Mothers

Community people mostly depend on the kabiraj, village doctors (quack) and traditional birth attendant during maternal complication. They normally do not refer the mother until a serious condition occurs as they know the barriers of communication and transportation during floods. Moreover, doctors are not available at healthcare facilities. The traditional practitioners refer the complicated mothers when they fail to provide necessary support. Sometimes they have to move from village doctor to Union sub-center, then from there to the Upazila health complex by boat, and then to the district hospital by any mode of transportation, which is very time consuming and painful for the mother.

“*My sister-in-law was admitted to Upazila hospital during the last flood with complication and she was referred to the Sylhet district hospital which took a minimum of five hours to reach by boat. Moreover, the wave of the river was too high to move and there was no other way to move and ultimately she delivered on the boat under high waves with high risk*”.—(P2, FGD 1)

“*Actually, in case of advanced pregnancy, women are in more trouble during the flood. There is no proper treatment of the mother during that period. The financial problem is common at that time for proper checkup and treatment of the mothers. Moreover, moving from the Khaliajhuri to another destination by boat is the only vehicle that takes lots of time and any accidents can happen at that time*”. —(P3, FGD 2)

“*If the traditional birth attendants cannot handle the complicated delivery then she refers the mother to the village doctor, and if village the doctor failed then he refers to the hospital. But before that, both of them tried traditionally. It takes a long time to decide to refer the mother to a healthcare facility. And after that, the transportation by boat takes a long time. So, as a whole we lose long time, resulting in more life-threatening complications for the mother*”.—(IDI7) One of the male guardians

### 3.5. Influence of Maternal Death by Natural Disaster

Most of the community people perceived that maternal deaths are seriously influenced by natural disasters like floods in the affected areas. As the mother, who has been already suffering from malnutrition and anemia with other diseases, cannot receive proper maternal care during flood, if any delivery complication arises, and the result is her maternal death during a natural disaster. Tarred decision and transportation delays are common during floods, which influence maternal death. Ignorance of traditional practitioners to identify risky mothers and negligence of the community people cause delays in decision making. Moreover, arranging a boat, managing money, selecting people to assist, and delay in boat arrival cause further delays in transportation.

“*Two years ago, during the flood, a daughter of my brother-in-law died with delivery complication in boat on the way from Khaliajhuri. At first, she was carried to the village doctor’s chamber which was too far away from her home but it was very difficult to find the said doctor at night. Then he suggested for going to Upazila hospital for severe complications. Eventually, she died on the way to hospital*”.—(P4, FGD1)

“*Mother can survive luckily with maternal complication during flood. But it is a very risk of maternal death. A mother and her infant died during the last flood period inside the boat while going to hospital with adverse weather and could not reach to hospital on time*”.—(P3, FGD2)

“*I madly swim for searching a vehicle to transfer my niece with maternal complications. But I could not manage the boat easily at night. At last, we started to move to Sylhet for receiving her treatment at night with stormy weather. However, later she died after delivering on the boat*”.—(P9, FGD 4)

“*During flood, the wave of the river is large enough and it is a risk of drowning even for a large-sized boat. So, community people afraid to go out from home with such stormy weather. Risky referral mother can’t transfer quickly to Mymensingh medical and Dhaka through water even in the emergency*”.—(IDI4), Male guardian

“*A maternal death occurred after four to five hours of delivery. The bleeding started immediately after delivery but it needs two hours to manage a boat. When the boat was arranged the mother died inside the boat with profuse bleeding*”.—(IDI9), One of the village doctors

### 3.6. Community Recommendation to Prevent Maternal Death Related to Disaster

The majority of participants mentioned about early transferring the pregnant women with complications to the referral hospitals. Many of them recommended improvement in the communication and transportation systems in these areas where natural disaster like flood is very common. Some people also recommended water ambulances for referring the risky mothers during disaster. Some people also suggested the establishment of temporary health camps for proper care of pregnant women during the flood period.

“*If complicated pregnant mother could be admitted to hospital before delivery, then many mothers’ life can be saved. If there is availability of qualified doctors in the nearby facilities during flood then maternal death can be prevented*”.—(P9, FGD5) One of the participants

“*if the government takes initiative for quick transfer of high-risk pregnant mother then there will be no maternal death would occur, as earlier it happened in this area*”.—(IDI18) One of the village doctors

## 4. Discussion 

Flood disaster occurs almost every year at the Khaliajhuri Upazila of the Netrakona district in Bangladesh. This disaster occurs immediately after the rainy season. The suffering lasts over the year. The flood intensity and severity are increasing over the years mainly due to climate change. During the flood period, there is difficulty in communication, transportation, housing, and getting safe water and food. Maternal- and child-care are most affected. Maternal deaths commonly occur during the flood period in the absence of proper care and treatment. Boats are the only mode of transport for transferring a referred mother to hospital which is risky, and time- and money-consuming. Many maternal deaths occur on the boats. Delays in decision-making and transport are common during the flood disaster period which influences and results in maternal deaths.

Flood is common during the rainy season every year in the study area, starting from the early rainy season till the next month of the season. Normally, 20%–25% of Bangladesh is inundated during the monsoon from June to September. In the case of extreme flood events, 40%–70% of the area can be inundated, which amply proves the extremity of flood events [18].

Pregnant women are identified as the most vulnerable human beings during floods in Bangladesh. Maternal deaths are common in this period with several complications like bleeding after delivery obstructed and prolonged labor. A high number of pregnant women are found affected by the floods, which is similar to the UNFPA findings where approximately 1.75% of the flood-affected mothers are pregnant in the nine districts of Bangladesh [7]. It has been noticed that pregnant women, children, the elderly, disabled people, and women are more vulnerable than the other sections of the population. During disaster, they are left behind in cases of emergency because they lack knowledge, mobility, and resources [8].

It is found that many mothers died during natural disaster due to lack of access to health facilities during flood and stormy periods. Many mothers refrain from using the toilet during the day and consequently suffer from urinary tract infections. Pregnant women, lactating mothers, and differentsly-abled women suffered the most. It is difficult to move before and after cyclones [19]. The rate of inadequate antenatal care increased from 1.3% to 3.9% during any disaster [2].

Maternal deaths are influenced by floods because delays in decision-making and transport are common when pregnant women exhibit complications during this period. The factors included delays in recognizing the problem and in decision-making to seek care; long distances to health facility; scarcity of money and/or unavailability of transportation, and the long duration of transportation by boat. In Nepal, it was found that 14% of pregnant women in transit to or from a facility during disasters died. Of those, 46% died in a public facility with maternal complications due to transport delay. This shows that more women are willing to reach healthcare facilities but transportation delays are causing death [20]. Another study shows that Pakistan has recently been gravely affected by the worst monsoon flooding in a century. The number of people directly affected by the floods stands at 20.2 million, with over 1.9 million houses reportedly damaged or destroyed and women and girls comprising 85% of the persons displaced by the floods [5]. Therefore, natural disasters are increasing in the Indian sub-continent, threatening more pregnant women due to climate change.

Pregnant women are seriously deprived of proper care and treatment during the disaster period. It is more difficult to refer pregnant women experiencing complications to proper treatment during this period. It is recommended from this study that flood shelters should have increased separate accommodations for pregnant women. At least one room should be earmarked for child delivery and infants [9]. A study found that serious threats to pregnant women and children (0–6 months) in flood-affected sub-centers were reduced by providing delivery kits to the Auxiliary Nurse Midwives, as this lowered the individual risk of being exposed to waterborne- and skin-diseases [11].

Community people recommended alerting people in authority of the need for special support for pregnant women during disaster and emergency management of transport of high-risk pregnant women. Some studies also support such suggestions for the welfare of the mothers and infants for healthy pregnancy and safe delivery during disasters [21].

## 5. Conclusions

Maternal deaths mostly occur during the rainy season in flood-affected areas. Negligence of maternal healthcare, unavailability of facilities and proper care services, dependency on unqualified doctors, communication and transportation problems, and barriers to referral of the pregnant women experiencing complications during floods cause maternal deaths. To prevent such unwanted maternal deaths during disaster, policy-makers need to take special initiatives including community awareness about preparations for maternal care, providing support for proper care and treatment by qualified service providers, and quick referral of the pregnant women experiencing complications to the hospitals. Special attention of people in authority is essential to decrease the decision-making and transport delays of the risky pregnant women to reach to the hospital from their homes and communities.

## Figures and Tables

**Table 1 ijerph-16-04594-t001:** List of participants in the qualitative study.

Qualitative Instruments	Age Range	Participants
Focus group discussions (FGD) (*n* = 3)	18–50 years	Neighbor of the deceased mother’s home includes Pregnant women, recently delivered mothers with their guardians, community group members, Elite person of the society Religious leader
In-depth interviews (IDI) (*n* = 8)	25–60 years	Female guardians (02) Male guardians (02) Traditional Birth Attendants (02) Village doctors (02)

**Table 2 ijerph-16-04594-t002:** Content of the focus group discussion and in-depth interview.

Area of Discussion	Types of Prompts Used
Perception on the occurrence of maternal death and natural disaster	Did any maternal deaths occur here? Why and when did the maternal deaths occur in this area? Were any natural disasters observed? When did the disasters occur here? What types of disaster occur here? Does the disaster occur regularly, and what is the duration of the disaster?
Practices of maternal healthcare during natural disaster	How is maternal healthcare provided during a disaster? What preparation is there during pregnancy and delivery complications at the time of a disaster? What do the community people do during complication at a disaster? Where do they go during maternal complications, and how?
Barrier of the marginalized community to practices on maternal care during disaster	What are the challenges for maternal care during a disaster? What are the types of obstacles faced during the referral of a risky mother? Was the referral of a mother ever delayed, and why?
Relation of maternal death with natural disaster	Is maternal death influenced by natural disaster? How does natural disaster cause maternal death (with example)?
Recommendation and initiatives to prevent maternal death during disaster	How can maternal deaths during natural disasters be prevented? What initiatives can be taken by the community people to overcome such a situation? What are the recommendations to increase maternal healthcare during a disaster?

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
