# Peer review of "Effects of Climate Change and Maternal Morality: Perspective from Case Studies in the Rural Area of Bangladesh"

_ijerph, 2019, doi:10.3390/ijerph16234594_

Round 1

Reviewer 1 Report

Thanks for choosing me as one of the potential reviewers. This manuscript is well written, however I have few minor comments as below:

Authors conducted 3 FGDs and several groups of population were included in those FGDS but did not clearly mention which FGD included which group of population. Better to mention as FGD needs a unique group of population. Authors recruited 2 research officers for conducting the FGDs and IDIs but did not mention what was the criteria/qualification of those researcher officers. In line 78, need to check the spelling "facilitating". It will be "facilitated". Authors did not mention how long it took to conduct each FGD and IDI. The study was conducted in only one sub-district, this should mention as a major limitation of the study.   As the study was conducted in one sub-district, the manuscript title might be changed to "Effects of climate change and maternal morality:
Perspective from case studies in selected rural area of Bangladesh" or something like that. Otherwise, it seems like a nationwide study.

Reviewer 2 Report

The study explored the community perception of maternal deaths influenced by natural disaster, practice of maternal complications during natural disaster among the rural population in Bangladesh. The topic is important as it might be an important aspect of reducing maternal mortality in the area. The following comments might help to improve the quality of the manuscript. 1. It seems ‘Flood’ is one of the major risk factors for influencing maternal death. Therefore, it should be reflected in the topic. 2. Is there evidence for the sentence “Due to climate change, Bangladesh is overexposed for natural 51 disasters such as floods.” The flood could be a consequence of poor vegetation. 3. How sample size was decided for this study? 4. For FGD, the participants should be homogeneous. However, authors stated that “FGD members consisted of the neighbors of the deceased mother’s family, male and female guardians of pregnant and recently delivered mothers, pregnant mothers, community group members, school teachers, religious leader, Union Parisad members and elite person of the society.” What kind of participants were involved in each FGD? Participants of each group should be clarified. 5. The number of participants in FGD seems a lot. Usually the number of participants range from 5 to 8. Otherwise, it will be hard for the facilitator to control. 6. Grammar problem in the sentence “During FGDs, one research officer facilitating the discussion whereas, other research officer took important notes” (2.2 data collection). Authors need to check language through the whole manuscript. 7. It is confused that authors described that they used both thematic and content analysis methods. Please clarify what method they actually used for qualitative data analysis. Is there theoretical framework? 8. What is traditional Ayurveda? Is kabiraj and village doctor government employed? It is expected to provide contextual information of service delivery of the study. 9. It is not surprised that the flood disaster will lead to poor perinatal care provision and more maternal mortality. Is there any specific figures to show the proportion of women died during flood season? 10. The value of the paper should be considered. Authors might need to emphasize on the barriers and service needs during flood and put forward the recommendations for policy making.
